# Functional and Patient-Centered Treatment Outcomes with Mandibular Overdentures Retained by Two Immediate or Conventionally Loaded Implants: A Randomized Clinical Trial

**DOI:** 10.3390/jcm10163477

**Published:** 2021-08-06

**Authors:** Javier Montero, Abraham Dib, Yasmina Guadilla, Javier Flores, Beatriz Pardal-Peláez, Norberto Quispe-López, Cristina Gómez-Polo

**Affiliations:** Department of Surgery, Faculty of Medicine, University of Salamanca, 37007 Salamanca, Spain; ibrahimdib@usal.es (A.D.); yguadilla@usal.es (Y.G.); j.flores@usal.es (J.F.); bpardal@usal.es (B.P.-P.); norbert_1404@hotmail.com (N.Q.-L.); crisgodent@hotmail.com (C.G.-P.)

**Keywords:** patient-reported outcomes, prosthodontics, dental implant, mastication, overdenture, immediate loading

## Abstract

This study aims to assess the treatment outcomes (functional and subjective) of mandibular overdentures retained on two implants with or without an immediate loading protocol. In this randomized clinical trial, twenty fully edentulous patients were treated with a mandibular two-implant-retained overdenture and a complete new maxillary denture. In half of the sample, the implants were loaded immediately by means of VulkanLoc^®^ abutments after emplacement of the implant, but in the counterparts, these VulkanLoc^®^ abutments were connected to implants two months after the surgery (conventional protocol), and until that time the dentures were retained by healing abutments. Treatment outcomes were assessed at two, six, and twelve months after surgery. Functional outcomes were calculated according to masticatory performance, estimated by the mixed fraction of a two-coloured chewing gum after five, ten, and fifteen chewing strokes, by the occlusal force recorded by pressure-sensitive sheets, and by the bioelectrical muscular activity. The subjective outcomes of the treatment were assessed using both the oral satisfaction scale (visual analogue scale) and the Spanish version of the Oral Health Impact Profile (OHIP-20). The findings of the present study show that new complete dentures resulted in significant improvements in chewing ability, patient satisfaction, and oral health-related quality of life and that subsequent implant-retained overdentures produced further and faster significant improvements. The loading protocol may influence those positive self-reported outcomes rather than the objective functional evaluations.

## 1. Introduction

European countries are witnessing a demographic evolution characterized by the aging of their populations. This increase in life expectancy has led to a disturbingly high prevalence of edentulism in Europeans aged 65 years or more; although in recent years in some countries there has been a positive trend towards a reduction in tooth loss among older adults [1]. Nevertheless, as people live longer there will be an increasing number of older people in the population, this group being the one most likely to need complete dentures.

In Spain, according to the last national oral health survey carried out in 2020 [2], between 4.4–10.2% of Spanish seniors (65–74%) were fully edentulous, and reported frequent disturbances of pain, eating restrictions, and a poorer quality of life [3] because of the poor functional performance of their conventional removable prostheses, mainly mandibular prostheses. For decades, default treatment for edentulous patients involved conventional dentures (complete removable maxillary and mandibular dentures dependent upon the residual alveolar ridge and mucosa for support and retention). Implant therapies have increased prosthodontics options for the treatment of edentulism. In 2002, in the “McGill Consensus Statement” [4], a panel of experts concluded there was overwhelming evidence that the restoration of edentulous mandibles with a conventional denture is no longer the most appropriate choice of prosthetic treatment with implant-supported overdentures (ISOD) becoming the care standard for edentulous mandibles. Later in 2009, the York Statement [5] concluded that a two-implant mandibular overdenture may be sufficient for most people, taking into account function, satisfaction, costs, and the clinical time involved.

The original protocol used in the emplacement of implants included a period without load (in which the implants were not involved in prosthetic retention) lasting between three and six months for the mandible and maxilla, respectively [6]. The waiting period obliges patients to remain in their previous situation of functional discomfort, delays the perception of therapeutic benefit, and may even affect the acceptance of the prosthetic treatment by the patient. Shortening of the restorative phase responds to the expectations of both patients and dentists. The concept of immediate loading is defined as the application of functional loads on the inserted implants along the 48 h immediately following emplacement in the bone [7]. One of the main criteria for immediate loading is to obtain high primary stability values, evaluated by means of insertion torque and ISQ [8,9]. High values of implant primary stability allow for a more predictable immediate loading [10].

Immediate loading may have some drawbacks since according to some authors early loaded implants may undergo micromotions at the anchoring site, leading to instability and fibrous encapsulation, hence preventing osseointegration and causing permanent implant mobility [11,12]. By contrast, other authors have reported that early physiological loading may promote peri-implant osteogenesis, the load acting as a stimulus of functional bone repair that leads to the fixing of the implant in the bone (osseointegration), as long as the functional loads are light or sufficiently moderate so as not to produce micromotions of more than 50 μm [13,14,15]. In a Cochrane review addressing randomized clinical trials, it was suggested that immediate or early loading is just as effective as conventional loading in properly selected patients with a sufficient amount and quality of mature bone with which to achieve high-torque insertion (>40 Ncm) [16].

Assessment of the impact generated by prostheses on quality of life is a relatively recent research line and few authors have addressed prosthodontic intervention in terms of oral health-related quality of life (OHQoL). Those who have done so have usually employed only a short-term follow-up (3–6 months) and a manifestly improvable quality design [17]. The evidence from randomized clinical studies suggests that implant-retained overdentures afford significantly better oral health (according to the OHIP questionnaire) than conventional complete dentures [4,5,18], although the magnitude of this benefit remains uncertain. Unfortunately, there is also a paucity of clinical studies aimed at quantitatively assessing the masticating function before and after prosthetic treatment [19] in order to check the functional change and the factors modulating this since the improvement of masticatory function is one of the chief wishes of patients requesting overdentures.

In 2008, Cannizzaro et al. carried out a randomized clinical study in which the biological and subjective results of mandibular overdentures with immediate loading were compared with conventionally loaded implants [20] The authors concluded that immediate loading seems to improve patient satisfaction without involving any risk associated with clinical parameters, as reported by others using a prospective design without randomization [21]. However, in these studies [20,21] the authors failed to perform objective tests of the masticatory yield of chewing force in each group that would allow such tests to be used as an external objective criterion against which subjective perceptions could be compared. Moreover, all overdentures were retained with bars that splinted the implants [20,21], even though there are less voluminous and less invasive elements that facilitate peri-implant hygiene, such as Locator^®^ abutments (Zest Anchor, Escondido, CA, USA) [22,23]. Moreover, the bar splinting the implants in the test group prevented patient masking. Thus, there is a paucity of data concerning immediately loaded unsplinted implants in overdenture patients, as well as of data on both the subjective and functional outcomes of mandibular overdentures. This randomized clinical trial was designed to test the null hypothesis that there would be no difference in clinical, functional, and patient-centered measures two, six, and twelve months after the emplacement of mandibular overdentures on two bilateral implants with either immediate loading or conventional loading.

The main aim of the present study was to assess the functional and subjective outcomes of mandibular overdentures retained on two implants with immediate loading as compared with implants with conventional loading for treating completely edentulous patients. The functional and subjective outcomes were assessed at two, six, and twelve months post-operatively.

## 2. Materials and Methods

This research was conducted in full accordance with ethical principles, including the World Medical Association Declaration of Helsinki (version, 2008 www.wma.net/, accessed on 1 July 2017). The experimental protocol was approved by the Bioethics Committee of the University of Salamanca (Spain) in 2015 (RUSAL_201500006834). All patients were informed about the aims, procedures, and duration of the study and were asked to provide written consent to participate in and undergo the implant interventions and prosthetic rehabilitation according to the standard guidelines of the Spanish General Council of Dentists. The data were acquired and analyzed according to current legislation relating to the privacy of personal data, clinical documentation, and biomedical research. In summary, we used aggregated data of the socio-demographic, clinical, functional, and subjective data of the patients, but no personal or affiliation information was included in the database.

This triple-blind randomized clinical trial (RCT) aimed to compare the clinical, functional, and subjective outcomes of patients who were initially fitted with new complete dentures and wore them for two months, after which the dentures were replaced by mandibular overdentures retained by two implants that were subjected to either immediate loading or conventional loading (functional load two months after implantation). The design of the study allowed two types of comparison to be made (Figure 1): cross-over (on comparing the intra-subject results at the beginning with new prostheses for two months and overdentures during twelve months) and also in parallel (comparing the results of overdentures with or without immediate loading). Since it would not be possible to have a control group (conventional complete prosthesis) blinded to the test treatment, we used pre-post controls (intra-subject comparisons before and after treatment).

Calculation of the sample size of this study is based on the primary result with the greatest dispersion according to our previous experience; i.e., the level of oral health-related quality of life according to the Spanish version of the OHIP [24], in whose validation the basal impact outcomes of 9 ± 8 points for patients bearing overdentures retained by Locator^®^ and of 14 ± 9 points for the bearers of conventional complete dentures have been established. Based on this dispersion detected in the reference population, and with the aim of detecting differences in the means of 5 points with two-sided tests with a power of 80% and an α error of 0.05, the number of patients required for each group was ten. Participants were recruited from patients seeking mandibular two-implant overdentures attending the Dental Clinic at the University of Salamanca.

Regarding the inclusion/exclusion criteria, all participants were completely edentulous individuals who had lacked teeth for more than ten years, routine users of conventional complete prostheses, with a sufficient amount of remaining bone to receive the implants with a minimum insertion torque of 40 Ncm, in the region of the mandibular canines (minimum height = 15 mm/minimum ridge width = 5 mm), with no evidence of systemic or psychic pathology that might contraindicate the implant treatment.

Functional data were collected by having the patients chew on pressure-sensitive colorimetric sheets (Dental PRESCALE, Fuji Photo Film Co., Tokyo; Japan) at maximum force while the muscular activity was recorded by surface electromyography (MYOMED_932 TM; Enraf-Nonius B.V., The Netherlands). The maximal voluntary occlusal force (Nw), the occlusal contact area (mm^2^), and the maximal/averaged pressure were evaluated for both the full arch and the anterior region by analyzing the degree of red colouring in the distinct areas with the pressure distribution mapping software (FPD8010 E, GC Corp., Tokyo, Japan). The density of the colour is proportional to the degree of pressure.

In addition, the masticatory performance was estimated by a mixing ability test of two-coloured chewing gums (Smint Kiss 3, Chupa Chups SL, Barcelona, España) as described by Montero et al. elsewhere [25] through the web https://studio.chewing.app/). The socio-demographic (age, gender, socio-occupational class) and subjective data (satisfaction, self-reported masticatory ability, and quality of life) were gathered in a face-to-face interview between the patient and the dentist.

In the pre-operative phase, the individuals who fulfilled the criteria for inclusion after the first screening by means of a clinical and radiological diagnosis were explored clinically to provide the basal anatomical picture with a series of clinical data (the shape of arches, height/width of the residual ridges, size of keratinized gingiva, together with the vertical dimension of occlusion, and transversal and sagittal maxilla–mandibular discrepancies). In the beginning, functional data were recorded with their current prosthesis according to Dental Prescale, surface electromyography on temporal and masseters at maximum force, and masticatory performance upon chewing two-coloured gum by mixing ability tests as described elsewhere [25]. Furthermore, the self-reported masticatory ability with respect to five target foods (apples, salads, meats, boiled vegetables, and carrots) according to the Leake Index [26] was also gathered. At the subjective level, data regarding oral satisfaction on the Visual Analogue Scale and OHQoL according to the Spanish version of the OHIP-20 (Oral Health Impact Profile) [24] were also collected.

After the initial screening, each patient received a new conventional maxillary and mandibular prosthesis following standard protocols (Prosthetic Phase I according to Figure 2). After the patients had worn the new prostheses for two months, the functional and subjective tests were repeated in all participants. These data were used as basal recordings of the standard treatment with conventional prostheses and allowed us to make intra-subject comparisons for the cross-over analysis.

After wearing the new conventional prostheses, all patients received local anaesthesia for the emplacement of two dental implants (Vulkan Internal Hex Implants, Barcelona, Spain) in the canine region, following standardized surgical protocols to emplace the implants in a crestal position with a minimum insertion torque of 40 Ncm. The primary stability values of the implants were recorded using a dynamometer (PCE-TM 80, Albacete, Spain) with a precision of ±1.5% and a measurement range from 0 to 147.1 Nw/cm in the clockwise and anti-clockwise directions. The quality of bone (from D1 to D4 according to the classification of Lekholm and Zarb) [27] and the spatial relationship of the inserted implants and the bone crest were recorded by both direct and X-ray inspection. In addition, the Implant Stability Quotient (ISQ) was recorded by Ostell^®^ at baseline and also at two, six, and twelve months) simultaneously with the peri-implant gingival index [28].

After implant placements, the implants received Locator^®^-like abutments named VulkanLoc^®^ (immediate loading group) or healing abutments (conventional loading group) whose height permitted an extramucosal exposure of at least 3 mm in height to give a certain degree of geometric retention of overdentures. To connect the mandibular prostheses, ball-headed burrs mounted on a hand-piece were used to grind down the internal surface of the prosthesis in contact with the abutments in order to emplace the complementary part of the VulkanLoc^®^ abutment with pink acrylic reliner (Kooliner GC, Kortrijk, Belgium) for the immediate loading group) or a silicon-based reliner to fit the contour to the healing abutment (GC Reline II soft, Kortrijk, Belgium) for the conventional loading group. The Teflon retention cap placed over the VulkanLoc^®^ abutment in this first phase had the least retentive power (400 grs: black). The overdentures were checked for occlusion to be properly bibalanced. Following this, a panoramic X-ray was taken in all of the patients with their acrylic prostheses set at maximum intercuspidation in order to obtain a radiographic record of the immediate post-operative period and thus monitor the basal position of the bone crest with respect to the implants for calculating vertical bone loss at the end of the follow-up (as depicts Figure 1). At seven days, the sutures were removed and the presence of pain, inflammation or wound dehiscence was recorded by inspection and palpation of the zone. From this time onwards, all consultations motivated by problems were attended to and recorded by the same investigators as those who participated in the implant phase and new denture phase, recording the occurrence of any biological or prosthetic complication.

After two months, all patients from both groups attended the “overdenture activation” session, proceeding as follows: in the patients of the group receiving immediate loading the black Teflon retention cap was replaced by one with greater retentive power (680 grs: blue), while in those receiving conventional loading the healing abutment was replaced by a VulkanLoc^®^ abutment, which was assembled directly onto a Teflon retention cap with the same retentive properties (680 grs: blue). One week later, clinical, functional, and subjective re-assessments were made. These evaluations of the clinical therapeutic result were repeated by the same investigators at six and twelve months, although in the latter revision a panoramic control X-ray was taken to allow the marginal bone loss of bone/year to be established, taking as reference the immediate post-operative situation (Figure 2).

Regarding patient-centered outcome measures, the self-reported oral satisfaction was collected on a visual analogue scale from 0 to 10 (OSS), with nominal extremes from “no satisfaction” to “full satisfaction” [29]. In addition, measurement of the impact on oral health-related quality of life (OHQoL) was accomplished using the Spanish version of the OHIP-20 [24], which is specifically designed to measure the quality of life of edentulous people across seven domains, i.e., functional limitation, pain, psychological discomfort, physical disability, psychological disability, social disability, and handicap. The OHIP questionnaires are descriptive measures of well-being in which individuals record the frequency of the appearance of oral impacts over the twelve preceding months (OHIP-PRE), or due to the prosthetic treatment (OHIP-POST), or how the individual thinks he or she was before the prosthetic treatment as he or she remembers (OHIP- THEN). The frequency responses were coded numerically from 0 = never to 4 = very often. The total scores of the questionnaires were calculated by counting the number of items with impact, that is, those perceived to occur occasionally or more frequently (≥2). Accordingly, the total score on the OHIP-20 ranges from 0 to 20.

The change in well-being was estimated by subtracting the total OHIP- POST score from the total OHIP-PRE score for the different times of observation. Theoretically, the patients who have perceived beneficial changes will obtain positive values and vice versa. Furthermore, these comparative scores were also calculated by subtracting the OHIP- POST from the OHIP-THEN in order to estimate the alpha change (the least biased estimation of the change). Finally, a retrospective global evaluation of changes in several aspects was collected by nine global transitional items (GTI) whose answers were much worse, worse, equal, better, or much better than before treatment.

### Data Analysis

The cross-sectional validity of the construct of the questionnaires was analyzed using the Pearson correlation coefficients between the pre-treatment scores (OHIP-PRE) and the initial oral satisfaction (on a scale of 0–10) and similarly at different observational moments. The longitudinal validity of the construct was evaluated by comparing the global oral satisfaction among groups who have perceived an overall change for the better and those reporting a worsening in at least one transitional item after overdenture treatments.

The validity of the cross-sectional construct of the questionnaires was confirmed by significant Pearson correlation coefficients between the pre-treatment scores (OHIP-PRE) and the initial satisfaction (r = −0.45; *p* < 0.05); between the post-treatment scores with complete denture (OHIP-POST1) and satisfaction with dentures (r = −0.45; *p* < 0.05); and between final post-treatment scores with overdentures and final satisfaction with overdentures (r = −0.83; *p* < 0.01). Similarly, the longitudinal validity of the construct was corroborated by comparing with Student *t*-test the global oral satisfaction between those patients who have perceived an overall change for the better (9.0 ± 1.0) with those who perceived a worsening in at least one global transitional item (6.9 ± 1.3) after overdenture treatment (T = 3.89; *p* = 0.001).

One property that can be demanded of indicators of oral quality of life is the possibility of the observation of minimum changes that are perceived as beneficial by patients (responsiveness). To assess the sensitivity of the change and provide clinical interpretability to the scores on change (recorded upon subtracting the OHIP-POST from the OHIP-PRE scores) we used a method inspired in earlier studies employing the OHIP [30], which detect several types of change (alpha, beta, and gamma changes), as reported in the methodological reference publication [31]. To carry out this analysis, three types of self-evaluation were used: Pre, performed before the intervention (new conventional prosthesis and overdentures); Post, performed after the intervention; Then, performed after the intervention but referring to the state prior to it. With these Pre, Post and Then assessments, the appropriate statistical checks were conducted to rule out the possibility that an apparent change in well-being might have occurred (Response Shift), either due to structural changes in the dimensional construct (gamma change) or due to a recalibration of the subject’s perceptions (beta change). After ruling out the presence of these possible biases, the measurement of the change in well-being attributable to the therapeutic effect (alpha change) will be legitimized by subtracting the OHIP-POST from the OHIP-THEN scores. This method would be the “gold standard” for the assessment of responsiveness (sensitivity to detecting changes in a questionnaire) and apparent changes (beta and gamma changes). The magnitude of change was estimated by the effect size which is calculated by dividing the mean change in well-being (OHIP_THEN—OHIP_POST) by the standard deviation of the total OHIP-THEN score. We used the ranges proposed by Cohen: <0.2, no effect; 0.2–0.5, a slight effect; 0.5–0.8, a moderate effect; >0.8, a strong effect [32].

Comparisons of nominal variables between groups were made with Chi-square tests and comparisons within groups were made with McNemar Tests. Regarding quantitative variables, if the distribution of the data violates the principles of normality and homoscedasticity, non-parametric tests (Mann–Whitney) tests were used instead of parametric ones (Student *t*-tests) for intergroup comparisons, and Wilcoxon instead of paired *t*-tests for within-groups comparisons.

The factors modulating clinical outcomes, well-being, and chewing function were estimated with stepwise linear regression analyses. Since they are quantitative and longitudinal variables, the clinical parameters were compared between groups (unpaired *t*-tests) and in the same subject (paired *t*-test). These variables were later introduced as predictive variables in the regression models adopted. The clinical parameters (in particular, the loss of an implant or the appearance of surgical or prosthetic complications) were used as dependent variables in the analysis of treatment outcomes.

## 3. Results

### 3.1. Sample Description

The sample was comprised of twenty adult patients equally distributed by sex, aged 66.3 ± 9.1 years, mostly belonging to the low socio-occupational class (75%) living in the city of Salamanca, Spain (65%), brushing their teeth at least once a day (80%), and currently non-smoking (75%). The Chi^2^ test found that the proportion of patients brushing their teeth less than once a day was significantly higher among the immediate loading group than among the conventional loading group (Table 1). However, both groups were comparable with respect to the other socio-demographic and behavioural variables.

### 3.2. Baseline Prosthetic Status

The assessment of the pre-operatory prostheses showed that the maxillary denture was older (115.7 ± 131.3 months) than the mandibular denture (97.2 ± 113.4 months), and that 80% of the mandibular denture had not enough retention nor stability, although half of the sample had the occlusion properly bibalanced and with proper interocclusal freeway space (Table 2). No significant differences between the groups were found in this regard.

### 3.3. Baseline Anatomical Conditions

The assessment of the anatomical parameters of the jaws (Table 3) shows that most implants were placed on type II quality of bone (60.0%), covered by thin or medium-soft tissue (88%), with an average of 2.6 ± 2.2 mm of attached gingiva. The bone volume at implant sites was on average 16.4 ± 5.3 mm in length and 6.9 ± 2.1 mm in width (measured 3 mm below the crest edge). No significant differences between the groups were found.

The most frequently used implant size was 3.75 mm (57.5%) or 4.2 mm (25%) in diameter and 11.5 mm (55%) or 10 mm (35%) in length.

### 3.4. Change in the Occlusal Pattern after Prosthetic Treatments

Table 4 shows the occlusal recordings according to the PRESCALE parameters (i.e., contact area, average pressure, maximal pressure, and occlusal load) for both the full arch and the anterior region, at baseline and at different observation times.

The paired *t*-tests shown in Table 4 found significant differences between the preoperative total occlusal area (11.7 ± 10.0 mm^2^) and the total occlusal load (123.2 ± 90.8 Nw) after the treatment with new complete dentures whose parameters increased to 16.3 ± 12.4 mm^2^ and 186.5 ± 154.2 Nw, respectively. In addition, after new dentures, the average and the maximal pressure in the anterior area increased from 8.4 ± 3.9 MPa to 10.1 ± 2.5 MPa and from 23.4 ± 13.2 MPa to 30.1 ± 8.1 MPa. These trends were also observed within the immediate loading group but not within the conventional loading group.

When the occlusal parameters were recorded after being worn for two months the implant-retained overdenture, the comparison with baseline recordings showed taht all parameters significantly increased, except the area and load registered in the anterior region. Despite no significant change being found after six months of wearing overdentures (likely because of the high data dispersion), all of the parameters significantly increased one year after the treatment with overdentures in both groups (Table 4).

If the occlusal parameters taken as reference were those recorded after a new complete denture instead of baseline records, then no significant improvement was observed in any parameter after wearing implant-retained overdentures during either two or six months. However, one year after treatment both the total occlusal area and the total occlusal load were significantly higher than that recorded two months after new dentures in both groups.

Differences between both groups were only significantly observed in the pre-operative phase with regard to the contact area and the occlusal load, which was significantly higher for those allocated to the conventional loading group (16.7 ± 11.4 mm^2^ versus 6.6 ± 5.2 mm^2^; and 171.9 ± 96.2 Nw versus 74.5 ± 53.6 Nw for the full-arch recordings).

### 3.5. Change of the EMG Muscular Activity after Prosthetic Treatments

Table 5 shows the variations of the EMG recordings in both the masseter and the temporalis muscles after two subsequent prosthetic treatments, i.e., a new complete denture and two-implant-retained overdentures followed for two, six, and twelve months. The findings revealed that, in the whole sample, there is a significant increment after one year of treatment with overdentures with respect to baseline records, but not with respect to records with new dentures. The temporalis muscles achieved higher values in comparison with the masseters. A trend towards a certain muscular symmetry was observed in all of the observational moments, except for two months after new overdentures in the conventional loading group, where right-side recordings were higher than in their counterparts. In this moment, and in this group, the activity of the left masseter was significantly lower (6.3 ± 3.8 μv) than that recorded two months after a new complete denture (12.3 ± 10.0 μv). At six months of follow-up, significantly increased activity in the left masseters and the right temporalis muscles was only observed for the immediate loading group. One year after overdenture treatment, only among the immediate loading group was a significant increment in masseter activity observed with respect to new complete denture recordings (28.1 ± 20.3 μv versus 12.5 ± 9.6 μv on the right side; and 22.9 ± 16.4 μv versus 11.4 ± 10.1 μv on the left side).

### 3.6. Change in Chewing Ability by Leake Index

In general, the chewing ability estimated by the Leake Index [26] increased gradually from baseline and after treatment with a new complete denture and even more after treatment with implant overdentures (Table 6). After treatment with new dentures, a McNemar test showed difficulties in chewing did not change significantly for carrots (Chi^2^ = 4.0; df:2, *p* = 0.14); salads (Chi^2^ = 4.8; df:2, *p* = 0.09); meats (Chi^2^ = 6.6; df:3, *p* = 0.09), vegetables (Chi^2^ = 3.0; df:2, *p* = 0.22); or apples (Chi^2^ = 3.3; df:1, *p* = 0.65). However, the change in chewing difficulties between baseline and both at six months and twelve months after overdenture were highly significant after McNemar tests for carrots (Chi^2^ = 11.0; df:2, *p* = 0.004 and Chi^2^ = 13; df:2, *p* = 0.002, respectively); salads (Chi^2^ = 12.0; df:3, *p* = 0.007 and Chi^2^ = 16; df:3, *p* = 0.001, respectively); meats (Chi^2^ = 11.8; df:3, *p* = 0.008 and Chi^2^ = 15; df:3, *p* = 0.002, respectively); boiled vegetables (Chi^2^ = 7.7; df:3, *p* = 0.05 and Chi^2^ = 7.7; df:3, *p* = 0.05, respectively); and apples (Chi^2^ = 12.0; df:2, *p* = 0.003 and Chi^2^ = 13; df:2, *p* = 0.001, respectively).

After implants, change was mainly observed for apples, salads, carrots, and meats but not for boiled vegetables. The most discriminative foods after prosthetic treatment were salads and meats. These findings were similar among the loading group, although the number of pattern foods chewed without any difficulty one year after treatment was significantly higher among the immediate loading group (4.2 ± 1.1 pattern foods) than among the conventional loading group (2.5 ± 1.9 pattern foods). Table 6 shows that the speed of being able to easily chew the pattern foods was significantly greater among the immediate than the conventional loading groups.

### 3.7. Change in Self-Rated Satisfaction after Prosthetic Treatments

Focusing on changes in self-rated satisfaction, Table 7 demonstrates that both prosthetic treatments applied sequentially (new dentures and two-implant-retained overdentures) increased significantly global satisfaction, satisfaction with aesthetics, and satisfaction with mastication. In general, global satisfaction significantly increased from 4.1 ± 3.2 to 7.3 ± 2.5 after new denture treatment and reached 8.5 ± 3.4 one year after overdenture treatment.

Regarding aesthetic satisfaction, the immediate loading group perceived higher values than the conventional loading group, being statistically significant at two and six months after implants, but not at the final self-report.

Within the loading groups, the changes in satisfaction after new dentures were comparable between groups, but taking as a reference the new dentures values, only the immediate loading group significantly increased in aesthetic satisfaction at two and six months after implants. By contrast, only within the conventional loading group was a significant improvement observed for all the satisfaction domains one year after implants with respect to new denture values (Table 7).

### 3.8. Change in Oral Health-Related Quality of Life after Prosthetic Treatments

Regarding the impact of quality of life (Table 8), the baseline records indicated that the most affected dimensions were *physical disability* (3.3 ± 1.4), *pain* (2.8 ± 1.1), and *functional limitation* (2.6 ± 0.8). After new dentures, the main domains with impact were *physical disability* (2.1 ± 1.6) and *functional limitation* (2.0 ± 0.8), and one year after implant treatment were also *functional limitation* (1.0 ± 1.2) and *physical disability* (0.8 ± 1.3), as depicted in Table 8.

In addition, it has been demonstrated that both groups afford comparable values in all of the observational moments, and in general, a significant improvement was detected across all domains except *handicap* after treatment with new dentures, and in most domains (e.g., *functional limitation, pain, physical disability*, and *social disability*) after implants in comparison with the new denture values. Table 8 also shows that after new dentures, the impact on some dimensions remains stable (such as *psychological* and *social disabilities*, and *handicap* among the immediate loading group, and *functional limitation*, *physical disability*, and *handicap* among the conventional loading group).

The validity of the cross-sectional construct of the questionnaires was confirmed by the significant Pearson correlation coefficients between the pre-treatment scores (OHIP-PRE) and the initial satisfaction (r = −0.45; *p* < 0.05); between the post-treatment scores with complete denture (OHIP-POST1) and satisfaction with dentures (r = −0.45; *p* < 0.05); and between final post-treatment scores with overdentures and final satisfaction with overdentures (r = −0.83; *p* < 0.01). Similarly, the longitudinal validity of the construct was corroborated by comparing the Student *t*-test with global oral satisfaction between those patients who perceived a change for the better overall (9.0 ± 1.0) with those who perceived a worsening of at least one global transitional item (6.9 ± 1.3) after overdenture treatment (T= 3.89; *p* = 0.001).

In addition, since further analyses with paired *t*-tests confirmed that there were no significant changes between the baseline records (OHIP-PRE) and the retrospective estimation of baseline impact (OHIP-THEN), the alpha change in the quality of life across domains was calculated by subtracting the Post-scores from the Then-scores (Table 9).

Table 9 depicts the true changes in OHQoL according to the versions _Post and _Then of the OHIP-20 after new dentures and after an overdenture (alpha change). In general, it is evident that the major effect size after new dentures was perceived among the *pain* and *functional limitation* domains, and among the *pain* and *physical disability* domains after the implants. The effect size in the global OHQoL of a new denture was slight (0.4 ± 1.3), but this value significantly increased until 1.7 ± 1.2, which implies a very strong effect in the global OHQoL after an overdenture, according to Cohen’s [29] benchmarks. Furthermore, the changes in *pain* were significantly higher among the immediate loading group than their counterparts after a new denture. The same trends were observed for *psychological disability* and *handicap* after overdentures. The intra-subject comparisons revealed that the number of items without impact across domains significantly improved after the overdentures in comparison with the new dentures values, except for *handicap*.

### 3.9. Retrospective Evaluation of the Well-Being Change by Global Transitional Items

Table 10 demonstrates that a new denture is able to improve the nine domains assessed retrospectively by transitional items in the majority of patients, although 10–15% of patients perceived that chewing ability and feeding satisfaction could be worse after a new denture. A significantly higher proportion of patients from the immediate loading group (80%) perceived that their *social relations* improved after a new denture (Table 10) than their counterparts (40%).

With regard to the effect of two-implant-retained overdentures, Table 11 shows that most patients felt better or much better after this treatment in all of the domains, although one patient from the immediate loading group felt they were worse at *pronouncing words*, felt *painful mouth*, perceived less *mouth comfortability* and worse *chewing ability*. However, the patients belonging to the immediate loading group felt significantly better *chewing ability*, *feeding satisfaction,* and *social relations* than their counterparts.

The minimally important difference (MID) of the OHIP-20, based on the true alpha changes values, was 1.3 ± 7.5 points after a denture and 7.1 ± 8.2 after overdentures. These benchmarks stated the mean change score that could be expected by patients that perceived a little improvement by global transition items. We calculated the global transition score by summing the scores of the GTI, i.e., positive judgements (better = +1; much better = +2), equal = 0, or negative judgements (worse = −1; much worse = −2), dividing it by nine domains and multiplying it by 100 to have a percentage estimation. Later, those patients with a 0 to 100 score were considered as having a minor positive change and those with a 100 to 200 score were considered as having a great positive change. The MID is a subjectively significant difference from the patient’s point of view.

### 3.10. Changes in Masticatory Performance by Mixing Ability Tests

No difference was found regarding the masticatory performance assessed by mixing ability tests after new dentures or overdentures (Table 12). The mixing ability increases with the number of cycles but reaches similar levels with the pre-operatory prostheses, the new dentures, and even with overdentures, although with five cycles it tended to be significantly higher with overdentures (24.9% ± 15.4%) than with new dentures (18.6% ± 14.7%).

### 3.11. Predictors of Patient-Centered Treatment Outcomes

The regression models summarized in Table 13 demonstrate that some clinical and patient-based outcomes are somewhat interrelated. Regarding mastication, the regression models point to better masticatory performance at five cycles as the bone quality increases (OR: 8.8–22.5), and proportional to the final temporalis muscle EMG activity in the right side (OR: 0.1–0.7). This model is able to predict 72% of the variance of the masticatory performance (corrected R^2^ =0.72). With respect to chewing ability, the regression model indicates that the number of easily chewed foods (Leake Index) is proportional to the masticatory performance at fifteen cycles (OR: 0.02–0.08) and depends also on the experimental cohort, being the immediate loading group the best chewers (OR: 0.6–3.0). This model was also highly predictable (R^2^ = 0.55). Focusing strictly on the patient, the final global satisfaction depends inversely on the number of biological complications (dehiscences, mucositis, etc.) and directly on the masticatory performance at five cycles. Finally, the impact on quality of life at the end of the study was proportional to the number of biological complications (OR: 1.2–3.7) according to this highly predictable model (R^2^ = 0.63).

## 4. Discussion

This RCT is pertinent because, to our knowledge, the evidence regarding the behaviour of Locator^®^-like abutments with immediate loading is still scarce. The design of this study is the best for the unbiased assessment of the interventions. The primary purpose of randomizing patients into treatment arms was to prevent researchers and patients from influencing their assessments or their self-reported outcomes, respectively, due to awareness of the treatment option applied. Concealing the allocation sequence from both researchers and participants eliminated this significant source of bias. To effectively conceal the randomization sequence, we used sequentially numbered, sealed, opaque envelopes, following the SNOSE guidelines [33]. These envelopes were only opened when the implants had been emplaced at more than 40 Ncm (inclusion criterion). Furthermore, to ensure the blinding of patients, all implants were rehabilitated with transmucosal abutments (they are visible by patients) in the same phase of implant emplacement, which improved the previous retention of the conventional prosthesis. Thus, patients assigned to the immediate loading group received Locator^®^-like abutments (with a retention of approximately 400 g) while those assigned to the conventional loading group received healing abutments (which generate an indirect retention via friction between its parallel outer walls and the base of the prosthesis of approximately 100 g). However, what is perceived by the patient is that two abutments protrude through the gum and that because of this, the retention of their lower prosthesis improves.

Moreover, as suggested Harry (Harris) et al. [34], in this study we provided new complete dentures to all patients in order to assess this conventional intervention but also to eliminate the variability of the quality of the pre-existing dentures, which would affect the assessment of measuring change after implant treatment.

In addition, to ensure the blinding of the investigators in the different assessments, the clinical team (performing the surgical and prosthetic interventions) were independent of the research team, which participated in the clinical, functional, and subjective evaluations at the different times of observation, whose strategic distribution prevented the members of the research team from knowing which group the patients belonged to. Moreover, since the rehabilitating clinicians did not participate in the post-operative evaluations of the patients, it was not expected that bias would appear in the response to the change in well-being owing to the influence that the presence of the clinical operators might exert on the patients treated (*halo effect*). In fact, the patients were informed that we were looking for ways to improve the procedure, such that they must tell us honestly what they felt and explain what we needed to improve. The criterion for inclusion established that all of the patients wished to be treated with mandibular overdentures precisely in order to avoid the bias of confusion that patients unsatisfied with their assignation generated in an RCT with parallel groups. There is evidence to suggest that the patient’s choice has a greater influence on a successful outcome than the operator’s preference for a treatment modality [35].

The findings of the present study showed that despite both groups being comparable in terms of socio-demographic, conductual, and anatomical variables, the full occlusal area and total occlusal load recorded at baseline was significantly higher among the conventional loading group (Table 4). This difference was annulated after treatments, but there is no rationale to explain this asymmetric distribution after randomization.

Regarding the change in the occlusal pattern, this study found that the occlusal area and the occlusal loading improves gradually from old dentures to new dentures and from new dentures to two-implant-retained overdentures (Table 4), as stated recently by Iwaki et al., who also used the Dental PRESCALE to evaluate the masticatory function [36].

However, the present study found greater areas of occlusion (26.1 ± 15.7 mm^2^) than that reported by Baca et al. [37] (10.6 ± 3.6) mm^2^, Suzuki et al. [38] (10.3 ± 5.2) mm^2^, and Iwaki [36] (7.0 ± 4.3) mm^2^, after using Dental PRESCALE in a comparable group. These differences could be explained because those authors used a distinct pressure sensitivity sheet (type 50 H, 97 μm thick) rather than the one used in the present study (MS type for medium pressure, 110 μm thick). The pressure sensitivity of the MS monosheet ranges from 10 to 50 MPa, whereas the pressure sensitivity of the 50 H sheet ranges from 5 to 120 MPa. To our knowledge, these 50 H sheets and their analyzing computer (Occluzer FPD703 or FPD707) are not currently available from the manufacturer (Fuji Photo Film Corporation), thus it should be desirable to assess the equivalence between those pressure-sensitive systems in order to compare past studies with new studies, as recently reported by Shiga et al. [39].

The final occlusal load found in the present study, 292.7 ± 163.2 N, is lower than that reported by Baca [37], (416.3 ± 137.2) N, and by Suzuki [38], (342.1 ± 163.6) N, but greater than that reported by Iwaki [36], (157.9 ± 60.3) N, among comparable patients. Furthermore, our values are similar to that reported by Baca [37] on patients treated with full-arch-fixed implant-supported restorations (254.9 ± 116.4 N) and greater than that observed among conventional complete denture wearers (242.0 ± 125.3 N) by Suzuki [38].

We agree with the aforementioned studies [36,37,38] in that occlusal forces and occlusal areas are greater among patients treated with overdenture than conventional denture wearers, and in that large differences are observed in the occlusal and functional parameters both at baseline and after treatment [40]. The Dental PRESCALE system is able to measure occlusal forces with high accuracy within the range of 20–80 N, although their readings are greater than actual load with and averaged difference of 6.3 ± 10.6 N [41].

No significant difference was found between immediate and conventional loading groups regarding the maximal occlusal load after overdenture treatments, as reported by Komagamine [42].

Alternatively, the influence of implant treatment on the muscular activity measured by electromyography (EMG) is scarcely studied [43]. A recent meta-analysis performed by von der Gracht [43] estimated that EMG recordings increased between 1.1–3.2 μV after treatment edentulousness with overdentures. The same study confirmed the huge data disparity ranging between 50–637 μV before treatment to 78–703 μV after treatment with overdentures.

Giannakopoulos et al. [44] used the same EMG recorder (Noraxon^®^) as in the present study and also found a great data dispersion and a stable right muscular asymmetry. Nevertheless, in contrast to Giannakopoulos et al. [44], in the present study, the temporalis muscles achieved higher values in comparison with the masseters as had been previously observed by Van der Bilt [45]. This finding was also reported by da Silva et al. [46] who, in agreement with our results, observed that patients with mandibular implant overdentures and an upper full prosthesis showed more balanced muscular activity after 15 months when compared with the initial period. This may indicate an improvement of the functioning of the masticatory system or at least a less pathogenic condition for musculoskeletal disorders. Hence, likely, the most relevant finding could rely on the trends towards asymmetrical activity rather than the magnitude of the bioelectric tone of the elevators muscles, which may be indicative of the reorganization of the neuromuscular system accompanied by an enhanced contractile capacity of the elevator muscles.

It is noteworthy that among the conventional loading group a certain irregular muscular pattern two months after implant was observed (Table 5), likely because they had just received the more retentive Locator^®^ caps and it may have disturbed the precedent muscular balance. It seems intuitive that an unstable mandibular denture requires greater muscular effort for its retention and stabilization than a well-retained overdenture. This fact may contribute to the great data dispersion and the lack of some pre–post differences shown in Table 5.

In consonance with the systematic review recently published by Boven et al. [47], our results support that the improvement with implant-retained overdentures in the masticatory performance, bite force, and the occlusal pattern is higher than after treatment with new complete dentures. Patients perceived they were able to chew better (Table 7), had greater occlusal load distribution (Table 4), and were able to easily eat tougher foods (Table 6) after implant-retained overdenture treatment compared with new conventional dentures. Some authors found that this functional improvement may remain stable for a long period of time (ten years) [40]. However, no significant difference was found by mixing ability tests of two-coloured chewing gum, maybe because chewing gum is an inadequate test food for completely edentulous patients due to its stickiness. It should be taken into account that all patients had a complete conventional maxillary denture. Müller et al. reported similar findings in a comparable study that also used two-coloured chewing gums for assessing masticatory performance [48].

Regrettably, there is a lack of studies measuring the effect of implant loading on mastication [49]. In agreement with Schuster, there are no significant differences between the groups in terms of masticatory performance. In the present study, the objective evaluation of the mastication was based on mixing ability tests, while Schuster et al. used the Sieving Method with an artificial test food (Optocal) for 40 cycles. Similar findings were reported by Komagamine et al. using colour-changeable chewing gum and gummy jelly to determine the masticatory efficiency [42]. Future studies should use comminution tests rather than mixing ability tests for assessing masticatory performance in denture wearers, because as much as the overdenture is properly retained, the upper one (a conventional complete denture) would be stuck to the lower one in the first contact, given the texture and stickiness of the chewing gum, preventing comfortable mastication, unless the occlusal surfaces had been previously lubricated with glycerine or something similar. Furthermore, the degree of adherence is proportional to the occlusal area, which has been significantly increased after treatment (Table 4). Self-reported chewing ability increased significantly faster among the immediate loading group than among the conventional loading group (Table 6), but this fact was not checked by objective assessments of the masticatory efficiency.

Focusing on satisfaction, a clear improvement has been observed after both the new denture dental treatments and the implant-retained overdenture dental treatments, being significantly higher with the last option. Treating CD wearers with implant-retained overdentures led to obvious improvements in patients’ satisfaction with their oral status and the impact on oral health-related quality of life [47,50,51]. A recent systematic review of randomized and comparative clinical trials examined the treatment outcomes of mandibular implant-supported and conventional dentures in opposition to complete conventional maxillary dentures using OHIP questionnaires [52], and found a statistically significant difference in the total scores favouring the implant–overdenture group compared to the conventional group. However, they found that the treatment effect is mainly perceived among *psychological discomfort* and *psychological disability*, but we have checked that major alpha change occurs among the *pain* and *physical disability* domains (Table 9), as reported by Schuster et al. [49]. An earlier meta-analysis carried out by Emami et al. [53], which included studies until 2007, concluded that both patient satisfaction and OHQoL are better with mandibular implant overdentures, although the magnitude of the treatment effect estimated by effect size was lower (95%CI: 0.4–1.2) than that obtained in the present study for the global impact on OHQoL (95%CI: 1.1–2.2). As expected, the changes in quality of life are moderate after new dentures and very strong after overdentures (Table 10 and Table 11), although some concerns arise due to the fact that around 5–15% of patients were worse in some dimensions (pronouncing, chewing, or hygiene) after treatment (Table 10 and Table 11).

According to our results, the minimally important difference in the OHIP-20 score after implant-overdentures (7.1 ± 8.2 points) is in the 7–10 range as reported by Allen et al. after removable prosthetic treatment [54] using the same OHQoL instrument.

In summary, a considerable consensus has been established that implant-retained mandibular overdentures can offer significant advantages to edentulous patients compared with new complete dentures. However, there is a lack of studies that assess the impact of immediate loading on patient-reported outcomes and mastication [49] (with most studies focusing on implants conventionally loaded), and without assessing the impact of the inconvenience of a long healing period, which may be relevant especially among senior patients. In this regard, Omura, in 2016, concluded that an immediate loading protocol tends to improve OHRQoL of mandibular overdenture wearers sooner than that observed with a conventional loading protocol one year after implant treatments. [55] However, the baseline OHIP scores of the participants (Japanese seniors) were already low at baseline, and the instrument is designed to quantify solely negative impacts in OHQoL. Our results align with Schuster, who also detected better and faster improvements in terms of OHQoL assessed by the OHIP-20 [49] among the immediately loaded overdenture wearers. Schuster also found greater effect size in the global OHIP scores among the immediate loading group (ES = 1.8) than among the conventional loading group (ES = 0.9) one year after the overdenture treatment. The same trend was observed for satisfaction with chewing, with aesthetics, and in general (Table 7). The immediate loading increased these positive self-reported perceptions.

Finally, this study has demonstrated that the number of biological complications is linearly correlated with both satisfaction and the final quality of life, and the correlation between prosthetic and biological complications shown in Table 13 is very interesting. The loading protocol (immediate vs. conventional) was found to be a significant predictor only for chewing ability in these regression models (Table 13).

The main limitations of the present study are the short follow-up period and the small sample size, therefore, future efforts should be directed towards checking the findings reported here with larger sample size and longer follow-up periods.

## 5. Conclusions

The major findings from the study are that new complete dentures resulted in significant improvements in chewing ability, patient satisfaction, and oral health-related quality of life and that subsequent implant-retained overdentures produced further and faster significant improvements. The immediate loading may increase, or at least accelerate, the above-mentioned self-reported outcomes, but there are no objective assessments of the masticatory performance, the muscular activity, or the maximum bite force.

## Figures and Tables

**Figure 1 jcm-10-03477-f001:**
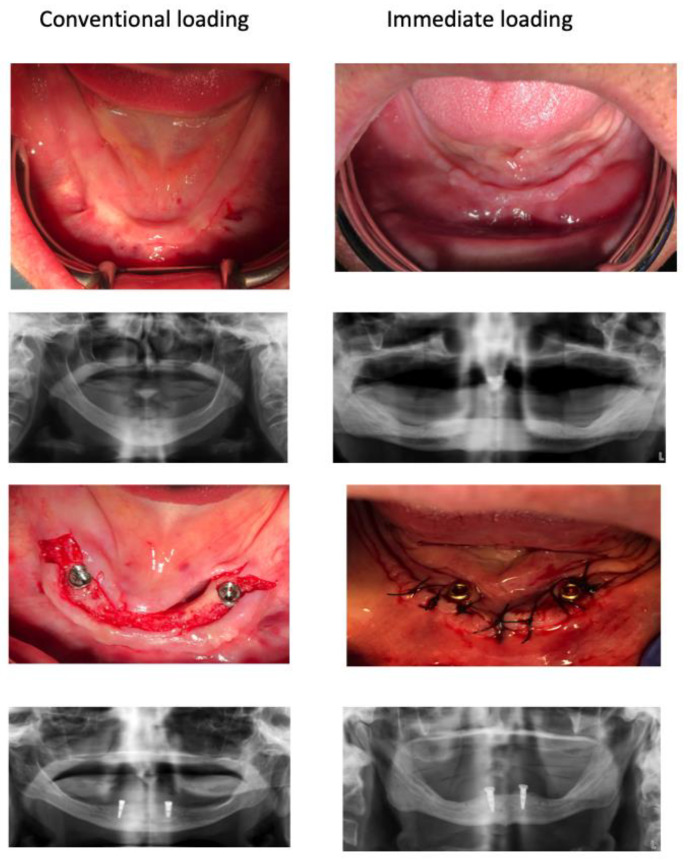
Pre-operative and immediate post-operative clinical and X-ray observations of a representative case of each group. After two months both groups were indistinguishable.

**Figure 2 jcm-10-03477-f002:**
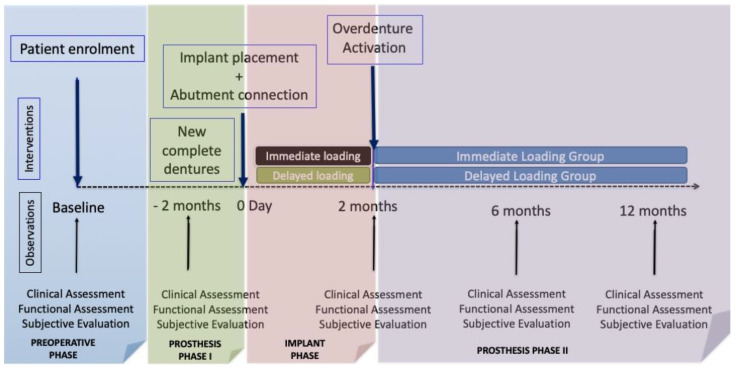
Diagram of the observations and interventions scheduled in the study.

**Table 1 jcm-10-03477-t001:** Socio-demographic and conductual description of the study sample (*n* = 20) and within implant groups.

SOCIO-DEMOGRAPHIC	ALL PATIENTS (*n* = 20)		Immediate Loading Group (*n* = 10)		Conventional Loading Group (*n* = 10)	
Age Interval	n	%	n	%	n	%
50–60 yrs	6	30.0	3	30.0	3	30.0
60–70 yrs	7	35.0	3	30.0	4	40.0
>70 yrs	7	35.0	4	40.0	3	30.0
	Mean	SD	Mean	SD	Mean	SD
Age (average)	66.3	9.1	67.6	9.6	65.0	8.9
Gender	n	%	n	%	n	%
Female	10	50.0	6	60.0	4	40.0
Male	10	50.0	4	40.0	6	60.0
Socio-occupational class	n	%	n	%	n	%
Low	15	75.0	8	80.0	7	70.0
Medium	5	25.0	2	20.0	3	30.0
Place of residence	n	%	n	%	n	%
Urban	13	65.0	6	60.0	7	70.0
Peri-urban	2	10.0	1	10.0	1	10.0
Rural	5	25.0	3	30.0	2	20.0
CONDUCTUAL						
Brushing habits *	n	%	n	%	n	%
2–3 times a day	8	40.0	4	40.0	4	40.0
Once a day	8	40.0	2	20.0	6	60.0
Less than once a day	4	20.0	4	40.0	0	0.0
Smoking habits	n	%	n	%	n	%
Non-smoker	10	50.0	5	50.0	5	50.0
Current smoker	5	25	2	20.0	3	30.0
Past Smoker	5	25	3	30.0	2	20.0
	Mean	Sd	Mean	SD	Mean	SD
Cigarettes/day among smokers	14.9	15.7	12.9	19.1	4.3	5.0
Pattern of dental check-ups	n	%	n	%	n	%
regularly	2	10.0	0	0	2	20.0
problem-based	18	90.0	10	100.0	8	80.0

* Chi^2^ test found significant differences between loading-based groups.

**Table 2 jcm-10-03477-t002:** Denture age and prosthetic status at baseline according to four parameters (retention, stability, interocclusal distance, and bibalanced occlusion) in the study sample (*n* = 20).

PROSTHETIC STATUS	ALL PATIENTS (*n* = 20)		Immediate Loading Group (*n* = 10)		Conventional Loading Group (*n* = 10)	
Denture age (months)	Mean	Sd	Mean	Sd	Mean	Sd
Maxillary Denture	115.7	131.3	135.5	126.7	93.7	140.3
Mandibular Denture	97.2	113.4	107.9	94.0	86.5	134.3
MANDIBULAR DENTURE QUALITY	n	%	n	%	n	%
Retention Does the denture dislodge with vertical pulling on central incisors after these are dried with gauze?						
No	4	20.0	3	30.0	1	10.0
Yes but difficult	4	20.0	2	20.0	2	20.0
Yes easily	12	60.0	5	50.0	7	70.0
Stability Is there movement induced by index and middle finger pressure on the first molar teeth?						
No	4	20.0	3	30.0	1	10.0
Yes but difficult	3	15.0	1	10.0	2	20.0
Yes easily	13	65.0	6	60.0	7	70.0
Interocclusal distance						
1–4 mm	9	55.0	4	40.0	5	50.0
<1 OR >4 mm	11	45.0	6	60.0	5	50.0
Bibalanced occlusion						
adequate	11	45.0	6	60.0	5	50.0
poor	9	55.0	4	40.0	5	50.0

**Table 3 jcm-10-03477-t003:** Anatomical baseline conditions of the jaws at the implant sites (*n* = 20), and distribution of the implant sizes. Comparisons by Chi-square and Student *t*-tests.

	ALL PATIENTS (*n* = 20)		Immediate Loading Group (*n* = 10)		Conventional Loading Group (*n* = 10)	
Bone Quality according to Leckholm & Zarb [24]						
	n	%	n	%	n	%
Type I	3	15.0	2	20.0	1	10.0
Type II	12	60.0	5	50.0	7	70.0
Type III	5	25.0	3	30.0	2	20.0
Averaged ridge dimensions at canine regions						
	Mean	SD	Mean	SD	Mean	SD
Length to basal bone	16.4	5.3	18.1	6.5	14.7	3.1
Witdth at 3 mm subcrestal	6.9	2.1	7.3	2.1	6.5	2.0
Gingival Biotype	n	%	n	%	n	%
fine	10	44.0	6	60.0	4	40.0
medium	9	44.0	3	30.0	6	60.0
thick	1	12.0	1	10.0	0	0
	Mean	Sd	Mean	Sd	Mean	Sd
Averaged attached gingiva at implant sites	2.6	2.2	2.9	2.6	2.4	1.8
Implant Size (*n* = 40)						
Diameter	n	%	n	%	n	%
3.3 mm	7	17.5	3	15.0	4	20.0
3.75 mm	23	57.5	13	65.0	10	50.0
4.2 mm	10	25.0	4	20.0	6	30.0
Length	n	%	n	%	n	%
10 mm	14	35.0	5	25.0	9	45.0
11.5 mm	22	55.0	13	65.0	9	45.0
13 mm	4	10.0	2	10.0	2	10.0

**Table 4 jcm-10-03477-t004:** Changes in the occlusal pattern according to the PRESCALE records after implant prosthetic treatments (*n* = 20).

BASELINE RECORDS	ALL PATIENTS (*n* = 20)		Immediate Loading Group (*n* = 10)		Conventional Loading Group (*n* = 10)	
FULL ARCH	Mean	SD	Mean	SD	Mean	SD
Contact Area (mm) *	11.7 ^a^	10.0	6.6 ^a^	5.2	16.7 ^a^	11.4
Average Preasure (MPa)	9.3 ^a^	3.8	8.4 ^a^	4.9	10.3 ^a^	2.2
Maximal Preasure (MPa)	30.3 ^a^	11.9	27.2 ^a^	15.8	33.4 ^a^	4.9
Occlusal Load (Nw) *	123.2 ^a^	90.8	74.5 ^a^	53.6	171.9 ^a^	96.2
ANTERIOR REGION	Mean	SD	Mean	SD	Mean	SD
Contact Area (mm)	5.1 ^a^	4.7	3.2 ^a^	3.5	6.9 ^a^	5.2
Average Preasure (MPa)	8.4 ^a^	3.9	7.0 ^a^	4.3	9.9 ^a^	2.8
Maximal Preasure (MPa)	23.4 ^a^	13.2	17.6 ^a^	14.0	29.2 ^a^	9.8
Occlusal Load (Nw) *	57.4 ^a^	57.5	38.1 ^a^	44.2	76.7 ^a^	64.8
POST-OPERATIVE RECORDS: NEW COMPLETE DENTURE						
FULL ARCH	Mean	SD	Mean	SD	Mean	SD
Contact Area (mm)	16.3 ^b A^	12.4	16.6 ^b A^	14.8	16.0 ^a A^	9.1
Average Preasure (MPa)	11.8 ^a A^	3.6	12.3 ^a A^	4.4	10.9 ^a A^	1.9
Maximal Preasure (MPa)	35.5 ^a A^	5.2	35.4 ^a A^	6.0	35.5 ^a A^	4.1
Occlusal Load (Nw)	186.5 ^b A^	154.2	198.7 ^b A^	194.2	168.2 ^a A^	74.8
ANTERIOR REGION	Mean	SD	Mean	SD	Mean	SD
Contact Area (mm)	6.5 ^a A^	7.8	6.4 ^a A^	9.5	6.7 ^a A^	5.1
Average Preasure (MPa)	10.1 ^b A^	2.5	9.6 ^b A^	2.2	10.8 ^a A^	2.9
Maximal Preasure (MPa)	30.1 ^b A^	8.1	28.0 ^b A^	9.3	33.1 ^a A^	4.9
Occlusal Load (Nw)	76.4 ^a A^	100.4	79.0 ^a A^	125.4	72.5 ^a A^	55.0
POST-OPERATIVE RECORDS: IMPLANT-RETAINED OVERDENTURE AFTER TWO MONTHS						
FULL ARCH	Mean	SD	Mean	SD	Mean	SD
Contact Area (mm)	21.6 ^b A^	21.3	29.6 ^b B^	27.9	13.6 ^a A^	6.6
Average Preasure (MPa)	10.6 ^b A^	1.7	10.3 ^b A^	1.7	10.9 ^a A^	1.7
Maximal Preasure (MPa)	35.3 ^b A^	4.7	35.8 ^b A^	4.6	34.7 ^a A^	5.0
Occlusal Load (Nw)	241.7 ^b A^	266.9	325.9 ^b A^	358.5	157.6 ^a A^	83.2
ANTERIOR REGION	Mean	SD	Mean	SD	Mean	SD
Contact Area (mm)	5.6 ^a A^	7.2	6.2 ^a A^	9.3	4.9 ^a A^	4.7
Average Preasure (MPa)	10.8 ^b A^	2.6	10.4 ^b A^	2.6	11.1 ^a A^	2.7
Maximal Preasure (MPa)	31.2 ^b A^	6.9	30.2 ^b A^	8.5	32.1 ^a A^	5.2
Occlusal Load (Nw)	68.7 ^a A^	93.5	78.6 ^a A^	122.6	58.8 ^a A^	57.8
POST-OPERATIVE RECORDS: IMPLANT-RETAINED OVERDENTURE AFTER SIX MONTHS						
FULL ARCH	Mean	SD	Mean	SD	Mean	SD
Contact Area (mm) *	17.6 ^a A^	12.1	27.3 ^b A^	16.1	12.1 ^a A^	3.9
Average Preasure (MPa)	14.1 ^a A^	7.5	10.7 ^a A^	1.2	16.1 ^a A^	9.0
Maximal Preasure (MPa)	57.4 ^a A^	68.4	37.2 ^a A^	3.8	68.9 ^a A^	85.9
Occlusal Load (Nw) *	205.2 ^a A^	142.0	311.3 ^b A^	196.9	144.6 ^a A^	49.4
ANTERIOR REGION	Mean	SD	Mean	SD	Mean	SD
Contact Area (mm)	6.9 ^a A^	6.3	9.0 ^a A^	9.5	5.6 ^a A^	4.2
Average Preasure (MPa)	12.9 ^a A^	5.3	10.3 ^a A^	1.1	14.3 ^a A^	6.3
Maximal Preasure (MPa)	32.7 ^a A^	5.7	34.2 ^b B^	6.9	31.8 ^a A^	5.3
Occlusal Load (Nw)	70.2 ^a A^	75.6	101.0 ^a A^	109.7	52.6 ^a A^	50.2
POST-OPERATIVE RECORDS: IMPLANT-RETAINED OVERDENTURE AFTER ONE YEAR						
FULL ARCH	Mean	SD	Mean	SD	Mean	SD
Contact Area (mm)	26.1 ^b B^	15.7	26.3 ^b B^	13.0	25.8 ^b B^	18.7
Average Preasure (MPa)	12.1 ^b A^	3.4	11.9 ^b A^	3.1	12.4 ^a A^	3.9
Maximal Preasure (MPa)	36.2 ^b A^	5.3	35.8 ^b A^	4.6	36.5 ^a A^	6.1
Occlusal Load (Nw)	292.7 ^b B^	163.2	296.4 ^b B^	111.1	288.9 ^b B^	209.4
ANTERIOR REGION	Mean	SD	Mean	SD	Mean	SD
Contact Area (mm)	9.4 ^b A^	7.6	8.2 ^b A^	7.8	10.6 ^b B^	7.5
Average Preasure (MPa)	10.5 ^b A^	1.9	10.2 ^b A^	1.5	10.7 ^a A^	2.4
Maximal Preasure (MPa)	32.1 ^b A^	6.4	31.3 ^b A^	3.3	32.9 ^a A^	8.7
Occlusal Load (Nw)	105.3 ^b A^	83.2	88.4 ^b A^	81.6	122.2 ^b B^	85.5

^a, b^ Lowercase distinct letters within the columns mean significant pre-post differences (*p* < 0.05) after paired *t*-test, which have the pre-operative values as reference. ^A, B^ Uppercase distinct letters within the columns mean significant pre-post differences (*p* < 0.05) after paired *t*-test, which have the values with new complete dentures as reference. * Significant differences *p* < 0.5 between groups after Student *t*-tests.

**Table 5 jcm-10-03477-t005:** Changes in the electromyographic records (μv) after prosthetic treatments (*n* = 20).

EMG at Maximal Force (μv)	ALL PATIENTS (*n* = 20)		Immediate Loading Group (*n* = 10)		Conventional Loading Group (*n* = 10)	
BASELINE	Mean	SD	Mean	SD	Mean	SD
Masseter right	14.2 ^a^	14.0	12.8 ^a^	10.3	15.5 ^a^	17.4
Masseter left	10.7 ^a^	9.5	10.1 ^a^	8.8	11.3 ^a^	10.5
Temporal right	18.6 ^a^	20.8	12.5 ^a^	11.9	24.6 ^a^	26.4
Temporal left	18.9 ^a^	22.5	19.7 ^a^	23.5	18.1 ^a^	22.7
NEW COMPLETE DENTURE						
Masseter right	16.4 ^a A^	14.9	12.5 ^a A^	9.6	19.9 ^a A^	18.3
Masseter left	11.9 ^a A^	9.8	11.4 ^a A^	10.1	12.3 ^a A^	10.0
Temporal right *	21.5 ^a A^	17.7	13.6 ^a A^	9.5	29.4 ^a A^	20.9
Temporal left	21.3 ^a A^	23.7	21.8 ^a A^	24.6	20.9 ^a A^	24.4
OVERDENTURE TWO MONTHS						
Masseter right	13.9 ^a A^	8.3	13.0 ^a A^	5.8	15.0 ^a A^	10.8
Masseter left *	9.7 ^a A^	8.6	12.7 ^a A^	10.7	6.3 ^a B^	3.8
Temporal right	22.5 ^a A^	16.5	18.9 ^a A^	10.4	26.5 ^a A^	21.6
Temporal left	16.5 ^a B^	14.4	16.0 ^a A^	11.6	17.0 ^a A^	17.8
OVERDENTURE SIX MONTHS						
Masseter right	23.3 ^b A^	21.9	21.1 ^a A^	19.4	26.0 ^a A^	25.9
Masseter left	22.1 ^b B^	20.4	25.5 ^b B^	23.2	17.9 ^a A^	16.8
Temporal right	33.9 ^b B^	29.4	24.0 ^b B^	17.5	46.3 ^a A^	37.3
Temporal left	28.5 ^a A^	31.0	26.3 ^b A^	24.4	31.3 ^a A^	39.3
OVERDENTURE ONE YEAR						
Masseter right	23.5 ^b A^	20.2	28.1 ^b B^	20.3	18.4 ^a A^	19.8
Masseter left	20.0 ^b A^	14.8	22.9 ^b B^	16.4	16.8 ^b A^	13.0
Temporal right	26.8 ^b A^	16.1	22.5 ^b A^	15.9	31.7 ^a A^	15.7
Temporal left	26.2 ^b A^	19.9	25.9 ^a A^	23.2	26.4 ^a A^	16.9

^a, b^ Lowercase distinct letters within the columns mean significant pre-post differences (*p* < 0.05) after paired *t*-test, taking the pre-operative values as reference. ^A, B^ Uppercase distinct letters within the columns mean significant pre-post differences (*p* < 0.05) after paired *t*-test, taking the values with new complete dentures as reference. * Significant differences *p* < 0.5 between groups after Student *t*-tests.

**Table 6 jcm-10-03477-t006:** Changes in chewing ability according to Leake Index after prosthetic treatments (*n* = 20).

ALL PATIENTS	CARROT		SALADS		MEAT		VEGETABLES		APPLE		Number of Foods Easily Chewed	
BASELINE ^a^	n	%	n	%	n	%	n	%	n	%	Mean	SD
Easy	2	10.0	1	5.0	1	5.0	9	45.0	0	0.0	0.7 ^a^	0.8
A bit difficult	2	10.0	6	30.0	5	25.0	4	20.0	3	15.0		
Very difficult	16	80.0	13	65.0	14	70.0	7	35.0	17	85.0		
Two months after new denture ^bA^												
Easy	3	15.0	4	20.0	4	20.0	14	70.0	0	0.0	1.3 ^bA^	1.0
A bit difficult	4	20.0	9	45.0	8	40.0	3	15.0	5	25.0		
Very difficult	13	65.0	7	35.0	8	40.0	3	15.0	15	75.0		
SIX months after overdenture ^bB^												
Easy	7	35.0	12	60.0	11	55.0	15	75.0	5	25.0	2.5 ^bB^	2.0
A bit difficult	8	40.0	4	20.0	5	25.0	3	15.0	9	45.0		
Very difficult	5	25.0	4	20.0	4	20.0	2	10.0	6	30.0		
Twelve months after overdenture ^bB^												
Easy	11	55.0	17	85.0	15	75.0	17	85.0	7	35.0	3.4 ^bB^	1.8
A bit difficult	6	30.0	1	5.0	3	15.0	1	5.0	9	45.0		
Very difficult	3	15.0	2	10.0	2	10.0	2	10.0	4	20.0		
**IMMEDIATE LOADING GROUP**												
BASELINE ^a^	n	%	n	%	n	%	n	%	n	%	Mean	SD
Easy	1	10.0	1	10.0	0	0.0	5	50.0	0	0.0	0.7 ^a^	0.8
A bit difficult	1	10.0	3	30.0	2	20.0	2	20.0	1	10.0		
Very difficult	8	80.0	6	60.0	8	80.0	3	30.0	9	90.0		
Two months after new denture ^bA^												
Easy	1	10.0	2	20.0	1	10.0	7	70.0	0	0.0	1.1 ^aA^	0.9
A bit difficult	2	20.0	5	50.0	5	50.0	2	20.0	2	20.0		
Very difficult	7	70.0	3	30.0	4	40.0	1	10.0	8	80.0		
Six months after overdenture ^bB^												
Easy	5	50.0	8	80.0	6	60.0	9	90.0	4	40.0	3.2 ^bB^	1.8
A bit difficult	4	40.0	1	10.0	3	30.0	1	10.0	5	50.0		
Very difficult	1	10.0	1	10.0	1	10.0	0	0.0	1	10.0		
Twelve months after overdenture * ^bB^												
Easy	7	70.0	10	100.0	9	90.0	10	100.0	6	60.0	4.2 ^bB^	1.1
A bit difficult	3	30.0	0	0.0	1	10.0	0	0.0	4	40.0		
Very difficult	0	0.0	0	0.0	0	0.0	0	0.0	0	0.0		
**CONVENTIONAL LOADING GROUP**												
BASELINE ^a^	n	%	n	%	n	%	n	%	n	%	Mean	SD
Easy	1	10.0	0	0.0	1	10.0	4	40.0	0	0.0	0.6 ^a^	0.7
A bit difficult	1	10.0	3	30.0	3	30.0	2	20.0	2	20.0		
Very difficult	8	80.0	7	70.0	6	60.0	4	40.0	8	80.0		
Two months after new denture ^bA^												
Easy	2	20.0	2	20.0	3	30.0	7	70.0	0	0.0	1.4 ^bA^	1.2
A bit difficult	2	20.0	4	40.0	3	30.0	1	10.0	3	30.0		
Very difficult	6	60.0	4	40.0	4	40.0	2	20.0	7	70.0		
Six months after overdenture ^bA^												
Easy	2	20.0	4	40.0	5	50.0	6	60.0	1	10.0	1.8 ^bA^	1.9
A bit difficult	4	40.0	3	30.0	2	20.0	2	20.0	4	40.0		
Very difficult	4	40.0	3	30.0	3	30.0	2	20.0	5	50.0		
Twelve months after overdenture * ^bB^												
Easy	4	40.0	7	70.0	6	60.0	7	70.0	1	10.0	2.5 ^bB^	1.9
A bit difficult	3	30.0	1	10.0	2	20.0	1	10.0	5	50.0		
Very difficult	3	30.0	2	20.0	2	20.0	2	20.0	4	40.0		

^a, b^ Lowercase distinct letters or ^A, B^ uppercase distinct letters within the columns mean significant differences (*p* < 0.05) after paired *t*-test or McNemar test, taking the pre-operative values or the complete denture values as reference, respectively. * Significant differences *p* < 0.5 between groups after Student *t*-tests.

**Table 7 jcm-10-03477-t007:** Changes in self-rated chewing, aesthetic, and global satisfaction in a 0–10 scale after prosthetic treatments (*n* = 20).

SATISFACTION (0–10 Range)	ALL PATIENTS (*n* = 20)		Immediate Loading Group (*n* = 10)		Conventional Loading Group (*n* = 10)	
PRE-OPERATIVE	Mean	SD	Mean	SD	Mean	SD
Global	4.1 ^a^	3.2	3.6 ^a^	4.1	4.6 ^a^	2.2
Aesthetic	5.1 ^a^	2.8	5.1 ^a^	3.8	5.0 ^a^	1.6
Chewing	3.4 ^a^	2.7	3.6 ^a^	2.7	3.1 ^a^	2.8
After New Dentures	Mean	SD	Mean	SD	Mean	SD
Global	7.3 ^bA^	2.5	7.8 ^bA^	3.0	6.8 ^bA^	1.9
Aesthetic *	7.9 ^bA^	1.9	8.7 ^bA^	1.4	7.1 ^bA^	2.0
Chewing	6.5 ^bA^	2.1	6.6 ^bA^	2.1	6.4 ^bA^	2.3
After Overdentures two months	Mean	SD	Mean	SD	Mean	SD
Global	7.7 ^bA^	2.2	8.0 ^bA^	2.5	7.3 ^bA^	2.0
Aesthetic *	8.6 ^bB^	1.5	9.3 ^bB^	1.1	7.9 ^bA^	1.5
Chewing	7.0 ^bA^	2.2	7.1 ^bA^	2.4	6.9 ^bA^	2.4
After Overdentures six months	Mean	SD	Mean	SD	Mean	SD
Global	8.5 ^bB^	1.6	8.9 ^bA^	1.2	8.1 ^bB^	2.0
Aesthetic *	8.7 ^bB^	1.3	9.4 ^bB^	1.0	8.1 ^bA^	1.2
Chewing	7.8 ^bB^	2.1	8.2 ^bB^	1.7	7.4 ^bA^	2.5
After Overdentures one year	Mean	SD	Mean	SD	Mean	SD
Global	8.5 ^bB^	1.4	8.5 ^bA^	1.6	8.5 ^bB^	1.3
Aesthetic	9.0 ^bB^	1.1	9.4 ^bA^	0.7	8.6 ^bB^	1.3
Chewing	8.0 ^bB^	1.9	8.1 ^bA^	2.3	8.0 ^bB^	1.6

^a, b^ Lowercase distinct letters or ^A, B^ uppercase distinct letters within the columns mean significant differences (*p* < 0.05) after paired *t*-test, taking the pre-operative values or the complete denture values as reference, respectively. * Significant differences *p* < 0.5 between groups after Student *t*-tests.

**Table 8 jcm-10-03477-t008:** Changes in oral health-related quality of life (OHIP-20), by simple count method ^φ^ after sequential prosthetic treatments, i.e., new dentures and overdentures (*n* = 20).

ORAL HEALTH-RELATED QUALITY OF LIFE OHIP-20	ALL PATIENTS (*n* = 20)		Immediate Loading Group (*n* = 10)		Conventional Loading Group (*n* = 10)	
PRE-OPERATIVE SCORES	Mean	SD	Mean	SD	Mean	SD
Functional limitation	2.6 ^a^	0.8	2.7 ^a^	0.5	2.4 ^a^	1.0
Pain	2.8 ^a^	1.1	2.8 ^a^	1.0	2.7 ^a^	1.3
Psychological Discomfort	1.6 ^a^	0.7	1.3 ^a^	0.7	1.8 ^a^	0.6
Physical Disability	3.3 ^a^	1.4	3.5 ^a^	1.1	3.0 ^a^	1.6
Psychological Disability	1.4 ^a^	0.9	1.1 ^a^	0.9	1.6 ^a^	0.8
Social Disability	1.4 ^a^	1.5	1.3 ^a^	1.4	1.5 ^a^	1.6
Handicap	0.9 ^a^	1.0	0.7 ^a^	1.0	1.0 ^a^	1.1
TOTAL	13.7 ^a^	5.1	13.4 ^a^	5.0	14.0 ^a^	5.4
NEW DENTURES SCORES	Mean	SD	Mean	SD	Mean	SD
Functional limitation	2.0 ^b A^	0.8	2.1 ^b A^	0.6	1.9 ^a A^	1.0
Pain	1.8 ^b A^	1.3	1.8 ^b A^	1.6	1.7 ^b A^	1.2
Psychological Discomfort	0.9 ^b A^	0.8	0.8 ^b A^	0.8	0.9 ^b A^	0.9
Physical Disability	2.1 ^b A^	1.6	1.6 ^b A^	1.7	2.6 ^a A^	1.5
Psychological Disability	0.5 ^b A^	0.8	0.6 ^a A^	1.0	0.3 ^b A^	0.7
Social Disability	0.6 ^b A^	1.2	0.9 ^a A^	1.5	0.2 ^b A^	0.6
Handicap	0.4 ^a A^	0.8	0.5 ^a A^	0.9	0.2 ^a A^	0.6
TOTAL	8.1 ^b A^	5.2	8.3 ^b A^	6.5	7.8 ^b A^	3.9
OVERDENTURES SCORES	Mean	SD	Mean	SD	Mean	SD
Functional limitation	1.0 ^b B^	1.2	1.1 ^b B^	1.1	0.9 ^b B^	1.3
Pain	0.6 ^b B^	0.9	0.5 ^b B^	1.0	0.6 ^b B^	0.8
Psychological Discomfort	0.5 ^b A^	0.8	0.2 ^b B^	0.4	0.7 ^b A^	1.0
Physical Disability	0.8 ^b B^	1.3	0.6 ^b A^	1.3	0.9 ^b B^	1.3
Psychological Disability	0.2 ^b A^	0.5	0.1 ^b A^	0.3	0.3 ^b A^	0.7
Social Disability	0.1 ^b B^	0.5	0.2 ^b A^	0.6	0.0 ^b A^	0.0
Handicap	0.2 ^b A^	0.5	0.2 ^a A^	0.6	0.1 ^b A^	0.3
TOTAL	3.2 ^b B^	4.1	2.9 ^b B^	4.7	3.5 ^b B^	3.6

^φ^ The presence of any impact among items was recorded as present if it was reported at the threshold of “occasional” or more frequently. The number of impacts per person was calculated by simple count of items with impact across domains. ^a, b^ Lowercase distinct letters or ^A, B^ uppercase distinct letters within the columns mean significant differences (*p* < 0.05) after paired *t*-test, taking the pre-operative values or the complete denture values as reference, respectively.

**Table 9 jcm-10-03477-t009:** Alpha changes (OHIP-THEN—OHIP-POST) in oral health-related quality of life and effect sizes ^φ^ after new dentures and overdentures (*n* = 20).

CHANGES IN ORAL HEALTH-RELATED QUALITY OF LIFE	ALL PATIENTS (*n* = 20)		Immediate Loading Group (*n* = 10)		Conventional Loading Group (*n* = 10)	
AFTER NEW DENTURES	Mean	SD	Mean	SD	Mean	SD
Functional limitation	0.5 ^a^	0.7	0.6 ^a^	0.5	0.4 ^a^	0.8
Pain *	0.6 ^a^	1.6	1.2 ^a^	1.1	0.0 ^a^	1.8
Psychological Discomfort	0.2 ^a^	1.1	0.4 ^a^	1.1	−0.1 ^a^	1.2
Physical Disability	0.2 ^a^	2.4	0.9 ^a^	2.6	−0.6 ^a^	2.0
Psychological Disability	0.2 ^a^	1.2	0.5 ^a^	1.3	−0.1 ^a^	1.0
Social Disability	0.3 ^a^	1.5	0.5 ^a^	2.0	0.0 ^a^	0.9
Handicap	0.3 ^a^	1.0	0.6 ^a^	1.1	0.0 ^a^	0.9
TOTAL	2.2 ^a^	7.7	4.7 ^a^	7.7	−0.4 ^a^	7.2
GLOBAL EFFECT SIZE ^φ^	0.4 ^a^	1.3	0.8 ^a^	1.2	0.1 ^a^	1.2
AFTER OVERDENTURES	Mean	SD	Mean	SD	Mean	SD
Functional limitation	1.6 ^b^	1.5	1.6 ^b^	1.2	1.5 ^b^	1.9
Pain	2.9 ^b^	1.8	3.0 ^b^	1.5	2.7 ^b^	2.1
Psychological Discomfort	1.0 ^b^	1.2	1.3 ^b^	0.8	0.7 ^a^	1.4
Physical Disability	2.3 ^b^	2.1	3.0 ^b^	1.6	1.5 ^b^	2.3
Psychological Disability *	0.9 ^b^	1.0	1.5 ^b^	0.9	0.3 ^b^	0.7
Social Disability	1.4 ^b^	1.4	1.7 ^b^	1.4	1.0 ^b^	1.4
Handicap *	0.6 ^a^	0.9	1.1 ^a^	1.0	0.1 ^a^	0.3
TOTAL	10.5 ^b^	7.8	13.2 ^b^	6.8	7.8 ^b^	8.2
GLOBAL EFFECT SIZE ^φ^	1.7 ^b^	1.2	2.1 ^b^	1.1	1.2 ^b^	1.3

^φ^ The effect size was calculated by dividing the global mean change in well-being (OHIP_THEN—OHIP_POST) by the standard deviation of the total OHIP-THEN score. The change can be interpreted as: <0.2, no effect; 0.2–0.5, a slight effect; 0.5–0.8, a moderate effect; >0.8, a strong effect, as suggested by Cohen [29]. ^a, b^ Lowercase distinct letters within the columns mean significant differences (*p* < 0.05) after paired *t*-test taking the change after new complete denture as reference. * Significant differences *p* < 0.5 between groups after Student *t*-tests.

**Table 10 jcm-10-03477-t010:** Retrospective evaluation of the prosthetic treatment (new denture) by global transition items in the whole sample (*n* = 20) and within loading groups.

EFFECT OF NEW DENTURES	Much Worse	Worse	Equal	Better	Much Better
ALL PATIENTS (*n* = 20)	n(%)	n(%)	n(%)	n(%)	n(%)
Pronouncing words	0(0.0)	0(0.0)	6(30.0)	11(55.0)	3(15.0)
Taste and smell	0(0.0)	0(0.0)	12(60.0)	8(40.0)	0(0.0)
Painful aching in the mouth	0(0.0)	2(10.0)	8(40.0)	8(40.0)	2(10.0)
Oral hygiene	0(0.0)	0(0.0)	9(45.0)	7(35.0)	4(20.0)
Chewing Ability	1(5.0)	2(10.0)	0(0.0)	11(55.0)	6(30.0)
Feeding satisfaction	1(5.0)	1(5.0)	2(10.0)	8(40.0)	8(40.0)
Mouth comfortability	0(0.0)	0(0.0)	5(25.0)	8(40.0)	7(35.0)
Appealing Smile	0(0.0)	0(0.0)	5(25.0)	7(35.0)	8(40.0)
Social relations *	0(0.0)	0(0.0)	8(40.0)	8(40.0)	4(20.0)
IMMEDIATE LOADING GROUP (*n* = 10)	n(%)	n(%)	n(%)	n(%)	n(%)
Pronouncing words	0(0.0)	0(0.0)	2(20.0)	6(60.0)	2(20.0)
Taste and smell	0(0.0)	0(0.0)	6(60.0)	4(40.0)	0(0.0)
Painful aching in the mouth	0(0.0)	1(10.0)	4(40.0)	4(40.0)	1(10.0)
Oral hygiene	0(0.0)	0(0.0)	4(40.0)	3(30.0)	3(30.0)
Chewing Ability	1(10.0)	1(10.0)	0(0.0)	4(40.0)	4(40.0)
Feeding satisfaction	1(10.0)	0(0.0)	1(10.0)	3(30.0)	5(50.0)
Mouth comfortability	0(0.0)	0(0.0)	1(10.0)	4(40.0)	5(50.0)
Appealing Smile	0(0.0)	0(0.0)	1(10.0)	4(40.0)	5(50.0)
Social relations *	0(0.0)	0(0.0)	2(20.0)	7(70.0)	1(10.0)
CONVENTIONAL LOADING GROUP (*n* = 10)	n(%)	n(%)	n(%)	n(%)	n(%)
Pronouncing words	0(0.0)	0(0.0)	4(40.0)	5(50.0)	1(10.0)
Taste and smell	0(0.0)	0(0.0)	6(60.0)	4(40.0)	0(0.0)
Painful aching in the mouth	0(0.0)	1(10.0)	4(40.0)	4(40.0)	1(10.0)
Oral hygiene	0(0.0)	0(0.0)	5(50.0)	4(40.0)	1(10.0)
Chewing Ability	0(0.0)	1(10.0)	0(0.0)	7(70.0)	2(20.0)
Feeding satisfaction	0(0.0)	1(10.0)	1(10.0)	5(50.0)	3(30.0)
Mouth comfortability	0(0.0)	0(0.0)	4(40.0)	4(40.0)	2(20.0)
Appealing Smile	0(0.0)	0(0.0)	4(40.0)	3(30.0)	3(30.0)
Social relations *	0(0.0)	0(0.0)	6(60.0)	1(10.0)	3(30.0)

* Significant differences *p* < 0.5 between groups after Chi-square tests.

**Table 11 jcm-10-03477-t011:** Retrospective evaluation of the prosthetic treatment (implant-retained overdentures) by global transition items in the whole sample (*n* = 20) and within loading groups.

EFFECT OF TWO-IMPLANT-RETAINED OVERDENTURES	Much Worse	Worse	Equal	Better	Much Better
ALL PATIENTS (*n* = 20)	n(%)	n(%)	n(%)	n(%)	n(%)
Pronouncing words	0(0.0)	1(5.0)	6(30.0)	8(40.0)	5(25.0)
Taste and smell	0(0.0)	0(0.0)	10(50.0)	8(40.0)	2(10.0)
Painful aching in the mouth	0(0.0)	1(5.0)	4(20.0)	11(55.0)	4(20.0)
Oral hygiene	0(0.0)	0(0.0)	9(45.0)	7(35.0)	4(20.0)
Chewing Ability *	0(0.0)	1(5.0)	0(0.0)	5(25.0)	14(70.0)
Feeding satisfaction *	0(0.0)	0(0.0)	2(10.0)	5(25.0)	13(65.0)
Mouth comfortability	0(0.0)	1(5.0)	1(5.0)	8(40.0)	10(50.0)
Appealing Smile	0(0.0)	0(0.0)	6(30.0)	10(50.0)	4(20.0)
Social relations *	0(0.0)	0(0.0)	9(45.0)	7(35.0)	4(20.0)
IMMEDIATE LOADING GROUP (*n* = 10)	n(%)	n(%)	n(%)	n(%)	n(%)
Pronouncing words	0(0.0)	1(10.0)	2(20.0)	3(30.0)	4(40.0)
Taste and smell	0(0.0)	0(0.0)	6(60.0)	2(20.0)	2(20.0)
Painful aching in the mouth	0(0.0)	1(10.0)	2(20.0)	5(50.0)	2(20.0)
Oral hygiene	0(0.0)	0(0.0)	3(30.0)	4(40.0)	3(30.0)
Chewing Ability *	0(0.0)	1(10.0)	0(0.0)	0(0.0)	9(90.0)
Feeding satisfaction *	0(0.0)	0(0.0)	1(10.0)	0(0.0)	9(90.0)
Mouth comfortability	0(0.0)	1(10.0)	0(0.0)	2(20.0)	7(70.0)
Appealing Smile	0(0.0)	0(0.0)	3(30.0)	5(50.0)	2(20.0)
Social relations *	0(0.0)	0(0.0)	2(20.0)	5(50.0)	3(30.0)
CONVENTIONAL LOADING GROUP (*n* = 10)	n(%)	n(%)	n(%)	n(%)	n(%)
Pronouncing words	0(0.0)	0(0.0)	4(40.0)	5(50.0)	1(10.0)
Taste and smell	0(0.0)	0(0.0)	4(40.0)	6(60.0)	0(0.0)
Painful aching in the mouth	0(0.0)	0(0.0)	2(20.0)	6(60.0)	2(20.0)
Oral hygiene	0(0.0)	0(0.0)	6(60.0)	3(30.0)	1(10.0)
Chewing Ability *	0(0.0)	0(0.0)	0(0.0)	5(50.0)	5(50.0)
Feeding satisfaction *	0(0.0)	0(0.0)	1(10.0)	5(50.0)	4(40.0)
Mouth comfortability	0(0.0)	0(0.0)	1(10.0)	6(60.0)	3(30.0)
Appealing Smile	0(0.0)	0(0.0)	3(30.0)	5(50.0)	2(20.0)
Social relations *	0(0.0)	0(0.0)	7(70.0)	2(20.0)	1(10.0)

* Significant differences *p* < 0.5 between groups after Chi-square tests.

**Table 12 jcm-10-03477-t012:** Changes in masticatory performance at five, ten, and fifteen chewing strokes assessed by mixing ability test of bicoloured chewing gum (*n* = 20).

MASTICATORY PERFORMANCE by Mixing Ability Tests https://studio.chewing.app/	ALL PATIENTS (*n* = 20)		Immediate Loading Group (*n* = 10)		Conventional Loading Group (*n* = 10)	
PRE-OPERATIVE SCORES	Mean	SD	Mean	SD	Mean	SD
5 cycles	17.0 ^a A^	16.4	19.9 ^a A^	17.7	13.8 ^a A^	15.3
10 cycles	32.6 ^a A^	14.3	33.5 ^a A^	13.4	31.6 ^a A^	16.1
15 cycles	53.7 ^a A^	23.2	48.2 ^a A^	23.0	59.8 ^a A^	23.4
After New dentures	Mean	SD	Mean	SD	Mean	SD
5 cycles	18.6 ^a A^	14.7	22.1 ^a A^	15.6	15.4 ^a A^	13.8
10 cycles	35.3 ^a A^	20.9	38.7 ^a A^	23.2	32.3 ^a A^	19.3
15 cycles	54.1 ^a A^	20.4	54.2 ^a A^	12.4	54.0 ^a A^	26.3
Two months after overdentures	Mean	SD	Mean	SD	Mean	SD
5 cycles	19.3 ^a A^	14.1	22.9 ^a A^	14.2	15.6 ^a A^	13.8
10 cycles	36.3 ^a A^	17.3	38.2 ^a A^	18.3	34.4 ^a A^	17.2
15 cycles	54.9 ^a A^	16.6	54.9 ^a A^	13.6	54.8 ^a A^	20.1
Six months after overdentures	Mean	SD	Mean	SD	Mean	SD
5 cycles	19.9 ^a A^	14.1	22.9 ^a A^	14.2	16.6 ^a A^	14.1
10 cycles	37.9 ^a A^	18.3	38.7 ^a A^	16.3	37.1 ^a A^	21.3
15 cycles	54.3 ^a A^	19.2	55.6 ^a A^	17.6	52.9 ^a A^	21.7
One year after overdentures	Mean	SD	Mean	SD	Mean	SD
5 cycles	24.9 ^a A^	15.4	25.1 ^a A^	11.3	24.8 ^a A^	18.7
10 cycles	35.2 ^a A^	20.7	35.0 ^a A^	19.6	35.4 ^a A^	22.7
15 cycles	54.1 ^a A^	20.8	56.3 ^a A^	17.5	52.4 ^a A^	24.0

^a^ Lowercase distinct letters or ^A^ uppercase distinct letters within the columns mean significant differences (*p* < 0.05) after paired *t*-test among five, ten, and fifteen cycles, taking the pre-operative values or the complete denture values as reference, respectively.

**Table 13 jcm-10-03477-t013:** Linear Regression Analyses for predicting the clinical and patient-centered treatment outcomes as a function of the age, sex, cohort, bone quality, final masticatory performance, final chewing ability, satisfaction, quality of life, final occlusal area, final occlusal load, and muscular activity (*n* = 20).

DEPENDENT Predictors	β	Error	T	*p*-Value	Lower CI95%	Upper CI95%
CHEWING PERFORMANCE 5 cycles ^a^						
Bone quality	15.6	3.0	5.2	0.001	8.8	22.5
Muscular activity in right Temporalis (μv)	0.4	0.2	2.4	0.04	0.1	0.7
NUMBER OF FOODS CHEWED EASILY ^b^						
Chewing performance at 15 cycles	0.05	0.01	3.6	0.002	0.02	0.08
Cohort (Conventional as reference)	1.8	0.6	3.3	0.005	0.6	3.0
GLOBAL SATISFACTION ^c^						
Number of biological complications	−1.4	0.4	−3.7	0.005	−0.5	−2.2
Chewing performance at 5 cycles	0.04	0.02	2.5	0.03	0.00	0.08
FINAL IMPACT ON QUALITY OL LIFE ^d^						
Number of biological complications	2.5	0.6	4.4	0.001	1.2	3.7

^a^ F = 15.1; *p* < 0.01. Corrected R^2^ = 0.72; ^b^ F = 12.0; *p* < 0.01. Corrected R^2^ = 0.55; ^c^ F = 8.5; *p* < 0.01. Corrected R^2^ = 0.58; ^d^ F = 19.4; *p* < 0.01. Corrected R^2^ = 0.63.

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
