# Peer review of "Functional and Patient-Centered Treatment Outcomes with Mandibular Overdentures Retained by Two Immediate or Conventionally Loaded Implants: A Randomized Clinical Trial"

_jcm, 2021, doi:10.3390/jcm10163477_

Round 1

Reviewer 1 Report

The manuscript entitled "Functional and patient-centered treatment outcomes with mandibular overdentures retained by two immediate or conventionally loaded implants. A randomized clinical trial" is an interesting RCT about the use of overdentures prosthesis on implant fixtures comparing immediate or conventional implant loading.

The study is very well described with a very thorough and detailed study design, the part of the methodology was treated brilliantly, the administration of the oral health related quality of life tests are very interesting to analyze the results.

I suggest some improvements:

- Introduction

Line 48-59 avoid inserting european denmark uk and spain in the text because otherwise the title must be contextualized. Stay generic with demographic citations.

Line 78-79 before immediate loading may ... insert the following sentence with the references "One of the main criteria for immediate loading is to obtain high primary stability values, evaluated by means of insertion torque and ISQ [PMID: 32098046 - https: // doi.org/10.3390/app10238623 - https://doi.org/10.3390/ma14020270]. High values of implant primary stability allow for a more predictable immediate loading [https://doi.org/10.1016/j.matpr. 2020.08.243].

Line 113-120 "Moreover…" this period should be included in the discussions, it is inappropriate in the introductory part

The null hypothesis of the study is missing at the end of the introduction part

- Methods

I recommend including the number and date of approval of the ethics committee in the text as well and not just at the end of the manuscript

I suggest detailing the inclusion and exclusion criteria for patients

How was the type of treatment chosen for the patients?
Where is the randomization part?
These topics are fundamental in the methods part

- Discussion

Line 556 - 586 Reduce the blinding part and make it more fluent for the reader;

delete the part of the inclusion criteria that need to be added in detail in the part of the methods

Streamline the final part of the discussion section (too long);

Please, include the limitations of the study

Insert part of Abbrevations at the end

I suggest to the authors to improve the manuscript strictly following the reviewers' suggestions.

Author Response

Dear reviewer:

Thank you very much for time and effort invested in your revision. We have addressed all your comments in order to improve the manuscript. The parts of the manuscript that have been modified according to your suggestions are highlighted in yellow throughout the text. A point-by-point response to your comments is detailed below consecutively. We hope all these modifications will be finally satisfactory. Warmest regards from authors

REVIEWER 1 COMMENTS

The manuscript entitled "Functional and patient-centered treatment outcomes with mandibular overdentures retained by two immediate or conventionally loaded implants. A randomized clinical trial" is an interesting RCT about the use of overdentures prosthesis on implant fixtures comparing immediate or conventional implant loading.

The study is very well described with a very thorough and detailed study design, the part of the methodology was treated brilliantly, the administration of the oral health related quality of life tests are very interesting to analyze the results.

I suggest some improvements:

Q1. Introduction

Line 48-59 avoid inserting european denmark uk and spain in the text because otherwise the title must be contextualized. Stay generic with demographic citations.

R1: Done, although the specific values of the last national oral health survey in Spain are pertinent to contextualize the topic.

Q2: Line 78-79 before immediate loading may ... insert the following sentence with the references "One of the main criteria for immediate loading is to obtain high primary stability values, evaluated by means of insertion torque and ISQ [PMID: 32098046 - https: // doi.org/10.3390/app10238623 - https://doi.org/10.3390/ma14020270]. High values of implant primary stability allow for a more predictable immediate loading [https://doi.org/10.1016/j.matpr. 2020.08.243].

R2: Done.

Q3. Line 113-120 "Moreover…" this period should be included in the discussions, it is inappropriate in the introductory part

R3. We think that this paragraph is pertinent, since it makes a reasoned criticism of the background  highlighting the weaknesses of previous studies, precisely to justify the design of the present study.  Hence this paragraph contains a review of the relevant literature that motivates the research question and the experimental design (which should be included in the Introduction section according to journal guidelines https://www.mdpi.com/journal/jcm/instructions )

Q4. The null hypothesis of the study is missing at the end of the introduction part

R4. This statement has now been included as suggested. “….there would be no difference in clinical, functional and patient-centred measures along the 2, 6 and 12 months after the emplacement of mandibular overdentures on two bilateral implants with either immediate loading or conventional loading”

- Methods

Q5. I recommend including the number and date of approval of the ethics committee in the text as well and not just at the end of the manuscript

R5. Done

Q6. I suggest detailing the inclusion and exclusion criteria for patients

R6. Done

Q7. How was the type of treatment chosen for the patients?
Where is the randomization part?
These topics are fundamental in the methods part

R7. It has been clarified in M&M that patients sought mandibular two-implant retained overdentures, and that randomization was performed by sealed opaque envelopes that were opened when implants were inserted at more that 40Ncm following SNOSE guidelines.  

Q8- Discussion

Line 556 - 586 Reduce the blinding part and make it more fluent for the reader;

R8.Done

Q9. delete the part of the inclusion criteria that need to be added in detail in the part of the methods

R9. Done

Q10. Streamline the final part of the discussion section (too long);

R10. Done

Q11. Please, include the limitations of the study

R11. Done

Q12. Insert part of Abbrevations at the end

R12. Done

Reviewer 2 Report

Materials and Methods:

  1. Please inform which implant brand it was used and the measures (length and diameter).
  2. There is no information about how and with which material the prostheses were reembased after implant placement. The type of static and dynamic occlusion should also be informed.
  3. Please add clinical pictures of both groups, including radiographs, at each stage of the treatment procedure. They are essential for a better understanding of the reader.

Discussion

4. Please inform if the biological and technical complications were also assessed (probably issue of another publication?).

5. Which were in general, the most complains and benefits reported by the patients after implant placement?

6. The disadvantages of placement of only 2 implants at the maxilla should be discussed.

Author Response

Dear reviewer:

Thank you very much for time and effort invested in your revision. We have addressed all your comments in order to improve the manuscript. The parts of the manuscript that have been modified according to your suggestions are highlighted in yellow throughout the text. A point-by-point response to your comments is detailed below consecutively. We hope all these modifications will be finally satisfactory. Warmest regards from authors

Reviewer 2 Comments

Comments and Suggestions for Authors

Materials and Methods:

Q1.Please inform which implant brand it was used and the measures (length and diameter).

R1. Done

Q2. There is no information about how and with which material the prostheses were reembased after implant placement. The type of static and dynamic occlusion should also be informed.

R2.Done

Q3. Please add clinical pictures of both groups, including radiographs, at each stage of the treatment procedure. They are essential for a better understanding of the reader.

R3. A new figure has been added to show both protocols in the initial phase, because after two months of treatment both groups were indistinguishable.

Discussion

Q4. Please inform if the biological and technical complications were also assessed (probably issue of another publication?).

R4. As you have wisely thought, this data collection is planned to be published in another paper.

Q5. Which were in general, the most complains and benefits reported by the patients after implant placement?

R5.The main complaints and benefits of a new denture and a new overdenture have been commented in Table 10 and Table 11 respectively.

Q6. The disadvantages of placement of only 2 implants at the maxilla should be discussed.

R6. All the implants were inserted in the lower jaw. Currently, only two-implant retained mandibular overdenture are still considered as the standard of care for full edentulous patients in terms of cost-effectiveness. This paper to contributes to support that statement.

Round 2

Reviewer 2 Report

All corrections and suggestions were performed.